# Would the Cephalic Development in the Purebred Arabian Horse and Its Crosses Indicate a Paedomorphic Process?

**DOI:** 10.3390/ani12223168

**Published:** 2022-11-16

**Authors:** Arcesio Salamanca-Carreño, Pere M. Parés-Casanova, Néstor Ismael Monroy-Ochoa, Mauricio Vélez-Terranova

**Affiliations:** 1Facultad de Medicina Veterinaria y Zootecnia, Universidad Cooperativa de Colombia, Villavicencio 500001, Colombia; 2Generalitat de Cataluña, 25798 La Seu d’Urgell, Catalonia, Spain; 3Facultad de Ciencias Agropecuarias, Universidad Nacional de Colombia, Palmira 763531, Colombia

**Keywords:** alloidism, celloid, nasofacial profile, gerontomorphy, neoteny

## Abstract

**Simple Summary:**

In a broad sense, “paedomorphosis” refers to the similarity within a species between the adult form and its juvenile counterpart, as seen in certain species such as dogs and cats. Yet in horse breeds, the study of phenotypic characteristics in juveniles primarily focuses on body and limb proportions, e.g., distal limb length. Conversely, in the Purebred Arabian Horse (PAH), the juvenile form is clearly representational in the adult, and in particular, the phenotypic characteristics of the head. In this study, we investigate paedomorphosis by examining the left lateral profile of the head from photographs of PAH and first-generation crossbred PAHs (F1 crossbreds). This enabled a somewhat consistent image to be examined. By utilizing methods of geometric morphometry, a set of 23 semi-landmarks were studied that defined the profile of the head. The results implied the concave profile of the juvenile remained consistent with age and, therefore, there is a retention of juvenile traits into adulthood. This suggests that PAHs and F1 crossbreds could be considered paedomorphic-like animals.

**Abstract:**

This study examined paedomorphosis in PAH and F1 crossbreds. A sample of 99 horses was selected from 40 different breeders and consisted of three groups: stallions (*n* = 16), mares (*n* = 53), and geldings (*n* = 30), ranging from 10 months to 27 years in age. All horses presented a concave celloid lateral left head profile in the acquired photographic images. The hypothesis proposed in this study suggested the lateral profile of the head in juveniles was representational in the adult form due to the neonate’s facial bones (part of the splanchnocranium) developing at a different rate to those of the skull. The methodology utilized geometric morphometrics to identify 23 landmarks so as to identify profile curvature indicative between the three groups (stallions, mares, and geldings). Principal component analysis reduced the number of variables to 14 examinable landmarks. Using a two-NPMANOVA and multivariate regression test, it was demonstrated that an isometric relationship between the concave celloid profile in the juvenile and its adult counterpart existed. This result supported the hypothesis that PAH and F1 crossbreds expressed a paedomorphic trait due to the adult form retaining the concave celloid profile identified in the juvenile.

## 1. Introduction

The notion of “paedomorphy” or “paedomorphosis” refers, in a broad sense, to the striking resemblance of the adult form, to its neonate or juvenile form in the same species [1,2]. This resemblance is undoubtedly of great ontogenetic interest, where delayed development leads to the adult retaining morphological features from its juvenile morphology [3]. Furthermore, paedomorphy has been described as a contributory factor to the process of domestication, implying relevance in the artificial selection criteria of breeding adults by humans [4,5].

Furthermore, paedomorphosis has been widely studied in many species of wild vertebrates, such as the shaved mouse or naked mole rat (*Heterocephalus glaber*) [6], southern short-tailed opossum (*Monodelphis dimidiata*) [7], and various amphibian species [8,9,10]. Comparatively, similar studies examining domestic animals are very scarce, whit dogs [11], rabbits [12], and horses [2,5,13] being the primary focus. In dogs, the development of the neonate and facial bones (which are part of the splanchnocranium) occur at differing rates to those bones of the skull, which are part of the neurocranium) [14]. Among horses, this is evident in the head of foals, which present a characteristic bulging of the forehead and a proportionally short face, which usually presents a very different profile in the adult forma [2,15]. In general, juvenilized phenotypic characteristics in horse breeds focus on body proportions [2]; for example, Falabellas and Shetlands ponies present larger heads with comparatively shorter limbs and differences in cranial proportions [2]. In the Purebred Arabian Horse (PAH), the visual perception of “juvenile” is very clear in the adult form with proportionally longer limbs and in the head, conservation of the characteristic concave profile seen in foals.

With this in mind, there is a scarcity of objective methods for quantifying skull profiles. In 1978, Dusek tried to quantify the curve of the naso-facial profile of the Kladrub horse (Evans and McGreevy, 2006) [16]. Dusek’s method assumes that the skull has a regular curve, something that seems unlikely. Evans and McGreevy (2006) calculated the area of the profile rostral to the concave [16], a method that has been used in subsequent research [17,18]. One advanced method for quantifying shape, although scarcely used for animal ethnology, is Geometric Morphometrics (GM), which allows the study of variation in size and shape separately, and how they can covariate with other variables [19].

GM is a method based on the Cartesian coordinates of referenced landmarks that preserves geometry configuration by allowing a statistical representation of real shapes. The technique is based on the use of homologous anatomical points (Cartesian coordinate landmarks *x* and *y*) to explore the morpho-space of objects. As it has been said, GM separates shape and size while allowing a visual representation of pure shape variation [20]. As with the new approaches to the study of biological problems, GM has increased its use exponentially in the last 10 years to answer biological questions in different areas, especially in integrative biology. Likewise, it has undergone constant changes and updates in its methodology, making the approach more accurate and able to solve increasingly complex problems. Considering the analytical advantages of GM, the purpose of our research is to evaluate developmental changes of the cephalic profile (alloidism) with age in the PAH and first-generation (F1) crossbred derivatives. To date, this research is unprecedented for PAH and F1 crossbreds and complements previous studies on this topic [21,22].

## 2. Materials and Methods

### 2.1. Sample

We used 99 original digital images of the head (16 stallions, 53 mares, and 30 geldings) from animals with an age range of 10 months to 27 years taken in left lateral profile form with a digital Nikon P530 42X Optical Zoom camera^®^. Animals belonged to the Arabian breed and its F1 crosses (Anglo-Arab, Spanish × Arabian, Arabian × Appaloosa…), all presenting a cephalic concave profile. Photographs included a 1-inch reference item to facilitate calibration by the software. Animals were from 40 different Spanish private breeders. Although any type of biometric measurement is a priori subject to possible precision errors of the method used, we did not intend to evaluate the measurement error, so only a photograph per individual was evaluated.

### 2.2. Geometric Morphometrics

The digital sample collected was studied using GM methods. A total of 23 reference *x*, *y* semi-landmarks were located on the dorsal profile of each head (left side, Figure 1) with the software TpsDig version 1.40 [23]. Semi-landmarks are in smooth curves, for which the exact location on the curve, which is estimated statistically, cannot be identified. The principle is to optimize the position of each point by allowing it to slide along the curve (the head profile) under study. After sliding the semi-landmarks, the configurations of all the coordinates were superimposed by the Generalized Analysis of Procrustes, standardizing the position, scale, and orientation of all the configurations.

First, a Principal Component Analysis (PCA) was performed to reduce the number of anatomical variables. Then, with the most informative semi-landmarks (discriminative values (>[0.2]), we proceeded to a two-way NPMANOVA (*Non-Parametric Multivariate Analysis of Variance*) from Euclidean distances and 9999 permutations to determine if there were differences between the three studied groups (stallions, mares, and geldings). Finally, multivariate regression was established using age (in months, logarithmically transformed) as an independent variable. The coefficient of determination models the quotient of the variances of the fitted values and observed values of the dependent variables (most informative semi-landmarks). To measure the deviations that occur within Wilks’ Lambda (λ) statistic, computed as the ratio of determinants, was used.

For all statistical analyses, we used MorphoJ v.1.07a [24] and PAST v. 2.17c [25]. The confidence level was established for all tests at 95%.

## 3. Results

### 3.1. Principal Component Analysis

The first two Principal Components (PC) explained 80.65% of the total variation observed (PC1 + PC2 = 66.60% + 14.05%), with 14 semi-landmarks finally chosen (Table 1). These semi-landmarks defined both the frontal and facial profiles (Figure 2).

### 3.2. Non-Parametric Multivariate Analysis of Variance

The two-way NPMANOVA test did not reflect statistically significant differences between the three groups studied (F = 0.012; *p* = 0.707) nor between the differences in ages (F = 0.015; *p* = 0.512) (Table 2).

### 3.3. Regression

Multivariate regression appeared to be statistically significant (R^2^ = 0.014; Wilks’ λ = 0.279; F_28,70_ = 6.46; *p(regr)* << 0.0001) (Figure 3). The R^2^ value of 0.014 means that the informative semi-landmarks were able to explain only 1% of the variation in returns. This result demonstrates no profile change with age.

## 4. Discussion

In numerous species of domestic animals, selection has often favored juvenile characteristics in the adult form. Head profiles and short snouts are a few examples of striking retention in features that promote infantile morphology and are considered an ancestral agriotype [26]. Changes involving this underdevelopment with respect to the ancestor are grouped into the category of “paedomorphy” [1,2,4,26]. Paedomorphic changes are also often related to changes in behavior [4,27,28]. During domestication, the tendency toward the selection of small-size individuals has been recorded, with mentions of the reduction in the skulls of cats, pigs, and other domestic animals [27,28]. These proportionally smaller features give the appearance of paedomorphy.

The head in particular is a region of great ethnic and functional importance, forming with the neck a biomechanical lever that acts during certain phases of locomotive gaits [29]. Even variations of the profile have been grouped under the generalization referred to as alloidism and further classified into straight or orthoid, convex or cirtoid, and concave or celloid [29], with the latter being relevant to PAH. According to Sañudo (2009), PAH presents small heads, “flat” by shortening of the face, sunken nape, small ears, and with prominent orbits; from a cranial perspective, the frontal head profile presents a short face and wide muzzle [29]. With reference to these traits in PAHs, and the similar comparative relationship of such traits in other domestic species, the appearance of paedomorphosis becomes more apparent in PAHs. However, this is not evident in other equine breeds, such as the Andalusian, Berber, Kladruber, and Clydesdale, where convex profiles (cirtoids) are noted, or the straight profiles (orthoids) of French Trotters [30]. There are some heavy draught horses (“coldbloods”), such as Bolognese, Percheron, Belgian Coldblood, or elipometrical (ponies) trait breeds, such as Haflinger, with a documented Arab influence, which present a concave celloid profile [30].

Such characteristics might imply that paedomorphy is associated with early domestication or appears as a morphological expression when other traits are selected, such as docility [27,28,29]. Aside from agricultural and transport requirements, other factors influencing selection might be deemed as visually pleasing, such as coat color or perceived “beauty”. Either way, each characteristic has had an important role in the formation of some breeds. In fact, in other domestic species, such as dogs, cats, and, more recently, rabbits, the selection by purely aesthetic parameters (sometimes even forgetting functional needs) has become evident, and selection for head aesthetics has become selection criteria in some species [27,29]. Hence, the findings in this study demonstrate the usefulness of GM in objectively measuring morphological characters of ethnological importance. Even though the photographic images could be deemed a limitation, the lateral consistency between images identified the relevant landmarks remained relatively consistent.

The results presented significant landmarks in PAH and F1 crossbreds indicative of the juvenile profile in the adult form. This was determined by the two-NPMANOVA test identifying the landmarks in the lateral profile of the head that remained constant between all three groups (stallion, mares, and geldings). This conclusion was supported by the multivariate regressions test where the development of the concave celloid profile reflected a clear isometry with age; that is, the absence of alloidic change when growing, and, therefore, a retainment of cephalic juvenile traits in adulthood, even after reaching sexual maturity. These results demonstrated a paedomorphy in PAH and F1 crossbreds manifesting as a typical concave celloid head profile of the juvenile into the adult stage. In our opinion, ancient breeds, such as the Arabian horse, would present paedomorphy rather as an ancestral retention than a morphological expression of psychological traits, such as docility.

## 5. Conclusions

The appearance of retained development represents the premise that paedomorphy largely defines many domesticated species.

This study examined the paedomorphy in PAH and F1 crossbreds in the left lateral profile of the head through photographic images between the juvenile form and its adult counterpart. Twenty-three landmarks were identified that were indicative between all three groups (stallions, mares, and geldings). These results utilized geometric morphometrics and tested the findings with a two-NPMANOVA and multivariate regression tests. The findings described a constant relationship between the concave celloid profile in the juvenile with that of the adult, supporting the hypothesis that PAH and F1 crossbred expressed a paedomorphic trait.

## Figures and Tables

**Figure 1 animals-12-03168-f001:**
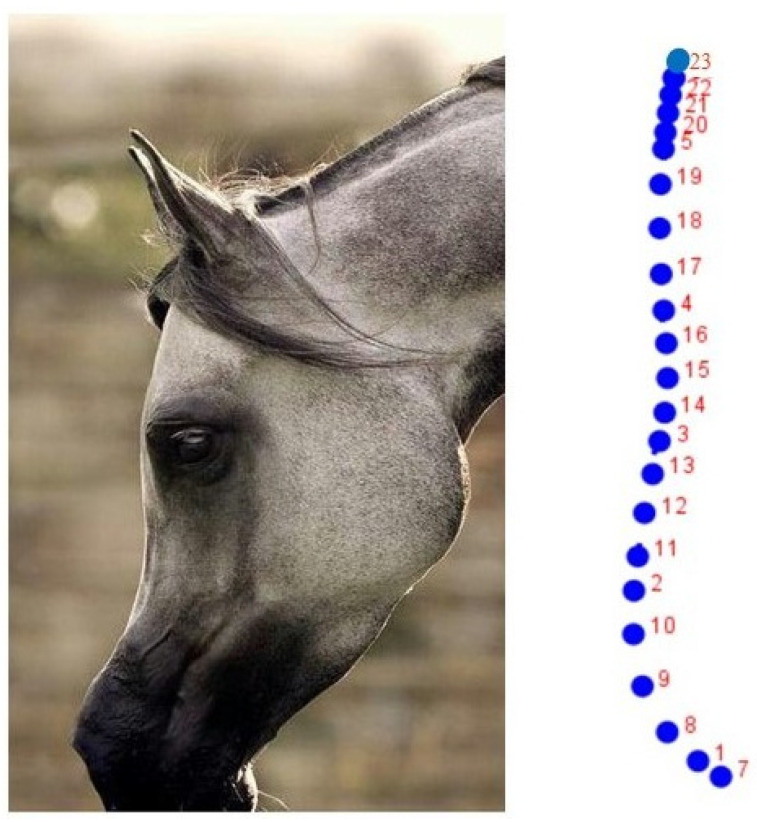
**Left**: Left lateral profile of the head. **Right**: Set of 23 semi-landmarks used initially to define the cephalic profile in the Purebred Arabian Horse. A sub-set of 14 semi-landmarks, which defined both the frontal and facial profile, was finally chosen (Table 1).

**Figure 2 animals-12-03168-f002:**
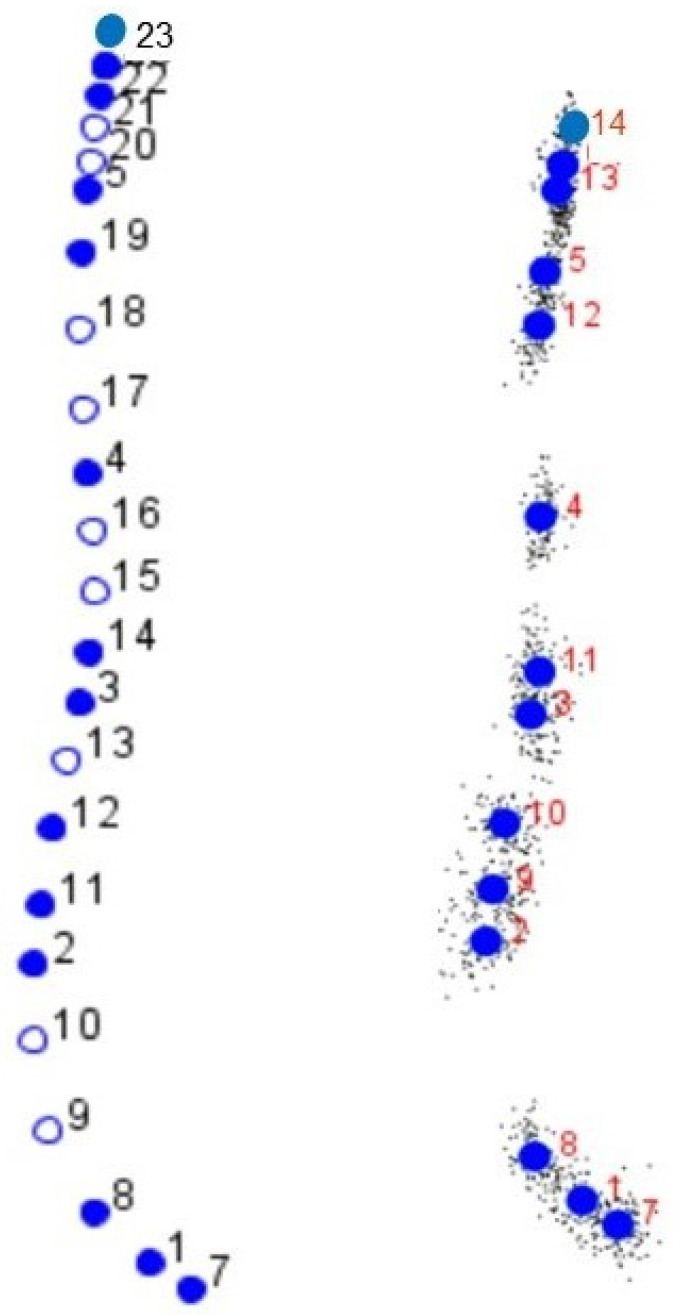
(**Left**): Set of the 14 semi-landmarks (empty dots) and semi-landmarks (filled dots), which were ultimately used for the study of the development of the cephalic profile in the Purebred Arabian horse and its F1 crosses. (**Right**): Semi-landmarks (filled dots) with small black dots on the right indicate the individual scattering for each semi-landmark.

**Figure 3 animals-12-03168-f003:**
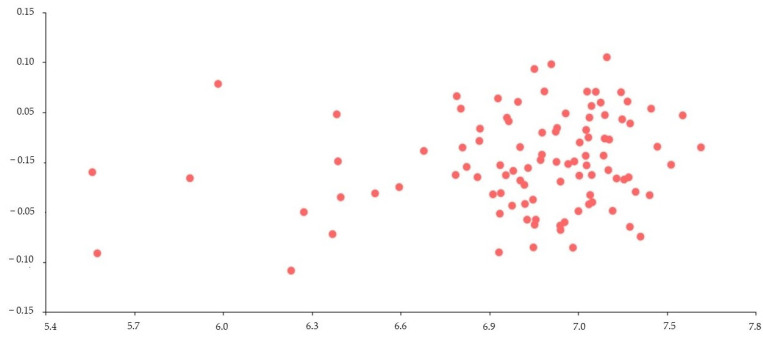
Multivariate regression using age in months (log-transformed values) as independent variable (*X*-axis) and the cephalic profile (defined by 14 semi-landmarks) as a dependent variable (*Y*-axis). Note: Each point represents an animal. A statistically significant correlation was reflected (R^2^ = 0.014; Wilks’ λ = 0.279; F_28,70_ = 6.46; *p*(*regr*) << 0.0001).

**Table 1 animals-12-03168-t001:** Results for Principal Component Analysis. The first two Principal Components (PC) accounted for 80.65% of the total variance observed (PC1 + PC2 = 66.60% + 14.05%). The 14 anatomical semi-landmarks with the highest discriminative value (>[0.2]) appear in bold.

	PC1 (66.60%)	PC2 (14.05%)
x1	**−0.2096**	−0.0583
y1	0.0995	**−0.3304**
x2	0.1513	0.0858
y2	0.0795	0.2615
x3	0.0527	0.0958
y3	**−0.2276**	0.1520
x4	−0.0181	−0.0060
y4	−0.1562	0.1029
x5	−0.0633	−0.0620
y5	−0.1545	**−0.3061**
x6	0.0232	−0.0250
y6	**0.3807**	0.0658
x7	−0.1659	−0.0309
y7	0.0063	**−0.3763**
x8	−0.1324	−0.1124
y8	0.1701	**−0.2313**
x9	0.0381	−0.1059
y9	0.1970	−0.0598
x10	0.1435	−0.0015
y10	0.1430	0.1217
x11	0.1431	0.1201
y11	0.0215	**0.2850**
x12	0.1141	0.1219
y12	−0.0736	**0.2288**
x13	0.0810	0.1320
y13	−0.1572	0.1948
x14	0.0432	0.0699
y14	−0.2104	0.1614
x15	0.0259	0.0261
y15	−0.1944	0.1480
x16	0.0038	0.0020
y16	−0.1746	0.1439
x17	−0.0339	−0.0006
y17	−0.1734	0.0376
x18	−0.0510	−0.0162
y18	−0.1584	−0.0862
x19	−0.0656	−0.0364
y19	−0.1499	**−0.2155**
x20	−0.0564	−0.0628
y20	−0.0351	**−0.2187**
x21	−0.0384	−0.0608
y21	0.1182	−0.1223
x22	−0.0092	−0.0505
y22	**0.2684**	−0.0241
x23	0.0239	−0.0244
y23	**0.3809**	0.0672

**Table 2 animals-12-03168-t002:** Two-way NPMANOVA test did not reflect statistically significant differences between the three sex groups studied (F = 0.012; *p* = 0.707) nor between the differences in ages (F = 0.015; *p* = 0.512).

Source	Sum of Squares	Degrees of Freedom	Mean Square	F	*p*
Sex	3.60 × 10^24^	2	1.80 × 10^24^	0.012022	0.7078
Age	7.31 × 10^25^	31	2.36 × 10^24^	0.015751	0.5212
Interaction	−3.14 × 10^26^	62	−5.06 × 10^24^	−0.033799	0.8221
Residual	4.49 × 10^26^	3	1.50 × 10^26^		
Total	2.12 × 10^26^	98			

## Data Availability

The data are available upon reasonable request to the second author.

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
