# Peer review of "Would the Cephalic Development in the Purebred Arabian Horse and Its Crosses Indicate a Paedomorphic Process?"

_animals, 2022, doi:10.3390/ani12223168_

Round 1
Reviewer 1 Report
The authors undertook to work out an interesting research problem, which is paedomorphism in Arabian horses and their crossbreeds. However, careful analysis of the manuscript may disappoint the reader. Research seems to be a continuation and addition to an article recently published by one of the authors, and quoted here. The research material used in both works is very similar and perhaps overlaps to a large extent. Unfortunately, in my opinion the submited paper does not describe the results exhaustively. According to Animals journal rules, manuscripts that are 'Article' type need to minimum word count of 4000 words. This paper does not reach even half of that value. The description of geometric morprometrics in the Introduction seems to be an unnecessary duplication from previous publications. The Results chapter is extremely laconic. The attached tables should rather be included in the supplement. Also, the Discussion is limited to four short paragraphs, does not refer sufficiently to the cited literature and requires considerable expansion.
In my opinion, the work in its current form cannot be published in "Animals". The manuscript requires a thorough revision and a detailed re-description of the results and their discussion.
I noticed a few minor editorial errors, e.g. inconsistent use of the term celloid / celoid, duplicate phrase "were taken" (line 17 or repetitions e.g. „morphology” (Line 42), „proces” (Lines43/44), „first” (Line 100) etc. Furthermore, the use of the word "skull" instead of cranium (neurocranium) is inappropriate.
Author Response
First reviewer’s responses
Dear reviewer
The authors appreciate the insightful comments.
We attach the all corrections and answers
Comment
“Research seems to be a continuation and addition to an article recently published by one of the authors, and quoted here”
Response
That article was on sexual dimorphism and it encompassed a smaller sample size
Comment
“The research material used in both works is very similar and perhaps overlaps to a large extent”.
Response
We agree, but GM applied to this kind of researches is very very scarce, so it is worthy to consider this techniques rather "novel" to the interested reader and thus be explained in detail.
Comment
In my opinion, the work in its current form cannot be published in "Animals". The manuscript requires a thorough revision and a detailed re-description of the results and their discussion.
Response
The manuscript is presented as a communication
Comment
I noticed a few minor editorial errors, e.g. inconsistent use of the term celloid / celoid, duplicate phrase "were taken" (line 17 or repetitions e.g. „morphology” (Line 42), „proces” (Lines43/44), „first” (Line 100) etc. Furthermore, the use of the word "cranium" instead of cranium (neurocranium) is inappropriate.
Response
All theses changes have been introduced.

Reviewer 2 Report
This study evaluated the ontogenetic allometry of the cephalic profile of purebred Arabian Horses and their crosses using geometric morphometrics, finding that it is paedomorphic. This finding is very interesting and valuable, but there are some concerns with the methodology. The comments follow.
There is a risk of significant measurement error due to the object angle against the camera lens. If the authors took multiple photographs per individual, they should evaluate the measurement error due to photograph differences. If not, the reliability of the results is questionable.
Why did the authors use the first two principal components, not the Procrustes coordinates, for response variables?
The ethics of animal use should be mentioned.
Lines 120-121: Why can the author say it is not significant though P < 0.0001?
Line 130: “CP” should be “PC”.
Line 131: What does it mean by “discriminant value“? I guess this table just shows loadings. What did the authors intend to suggest by bolding the highest “discriminant value”?
Figure 1: The definition of landmarks was not described. They should be clearly described.
Figure 3: The axis titles should be described.
Author Response
Second reviewer’s responses
Dear reviewer
The authors appreciate the insightful comments.
We attach the all corrections and answers
Comment
Why did the authors use the first two principal components, not the Procrustes coordinates, for response variables?
Response
All theses changes have been introduced.
Comment
The ethics of animal use should be mentioned.
Response
Corrected Review Board Statement: It was not necessary to receive the approval of an animal ethics committee, since in no case there was traumatic manipulations of animals or obtention of samples that caused pain. Moreover, there were no other risks for animals when conducting this research.
Comment
Lines 120-121: Why can the author say it is not significant though P < 0.0001?
Response
Regression analysis has been more explained and discussed. In the used test, the null hypothesis is the presence of a regression.
Comment
Line 130: “CP” should be “PC”.
Response
Corrected
Comment
Line 131: What does it mean by “discriminant value“? I guess this table just shows loadings. What did the authors intend to suggest by bolding the highest “discriminant value”?
Response
This is the common PCA procedure.
Comment
Figure 1: The definition of landmarks was not described. They should be clearly described.
Response
Semi-landmarks are defined on a curve, so it is not possible to locate them with anatomical precision.
Comment
Figure 3: The axis titles should be described.
Response
They appear on the foot paragraph

Round 2
Reviewer 1 Report
Unfortunately, despite some corrections by the authors, the manuscript still does not meet the criteria for Research Article in my opinion. While the still insufficient number of words or the number of References should not decide on the acceptance or rejection, the content of the Results chapter is limited only to the laconic presentation of two tables and one graph, without their discussion. The authors declare that this is just a communicate, but I did not notice it in the template. Besides, the results look promising, they only need to be solidly elaborated. So far, the authors have not coped with this. Therefore, I propose to withdraw the manuscript and its resubmission.
Author Response
Dear reviewer
The authors appreciate the insightful comments.
We attach the corrections and answers
Comment
Unfortunately, despite some corrections by the authors, the manuscript still does not meet the criteria for Research Article in my opinion. While the still insufficient number of words or the number of References should not decide on the acceptance or rejection, the content of the Results chapter is limited only to the laconic presentation of two tables and one graph, without their discussion. The authors declare that this is just a communicate, but I did not notice it in the template. Besides, the results look promising, they only need to be solidly elaborated. So far, the authors have not coped with this. Therefore, I propose to withdraw the manuscript and its resubmission.
Response
The manuscript has been corrected
The editor was told to change it to "communication"

Reviewer 2 Report
The authors partially revised the manuscript in accordance with the reviewers' comments, although they failed to address some points.
1. They did not respond to the following comment, "There is a risk of significant measurement error due to the object angle against the camera lens. If the authors took multiple photographs per individual, they should evaluate the measurement error due to photograph differences. If not, the reliability of the results is questionable." I hope the authors address this point carefully.
2. The authors stated, "Semi-landmarks are defined on a curve, so it is not possible to locate them with anatomical precision." as a response to the comment, "Figure 1: The definition of landmarks was not described. They should be clearly described." As the authors stated, semi-landmarks are located on curves, but landmarks are not. Landmarks can be and should be clearly defined and described.
Author Response
Dear reviewer
The authors appreciate the insightful comments.
We attach the corrections and answers
Second reviewer’s responses Round2
The authors partially revised the manuscript in accordance with the reviewers' comments, although they failed to address some points.
Comment
- They did not respond to the following comment, "There is a risk of significant measurement error due to the object angle against the camera lens. If the authors took multiple photographs per individual, they should evaluate the measurement error due to photograph differences. If not, the reliability of the results is questionable." I hope the authors address this point carefully.
Response
It has been added the following sentence: "As it was not intended to evaluate the measurement error, only a photograph per individual was evaluated. " is included in the text
- The authors stated, "Semi-landmarks are defined on a curve, so it is not possible to locate them with anatomical precision." as a response to the comment, "Figure 1: The definition of landmarks was not described. They should be clearly described." As the authors stated, semi-landmarks are located on curves, but landmarks are not. Landmarks can be and should be clearly defined and described.
Response
All points are semi-landmarks, as no perfect anatomical reference points (e.g., true landmarks) were used

Round 3
Reviewer 1 Report
I am pleased to state that the manuscript has been sufficiently improved to warrant publication in Animals. I hope the article will be of great interest to the readers.
Author Response
dear reviewer
Thank you for your appreciation